# Efficacy of vaccines based on chimeric or multiepitope antigens for protection against visceral leishmaniasis: A systematic review

**Karine Ferreira Lopes**[1]*, **Mariana Lourenço Freire**[2], **Silvane Maria Fonseca Murta**[1], **Edward Oliveira**[1]*

1 Genômica Funcional de Parasitos, Instituto René Rachou–Oswaldo Cruz Foundation, Belo Horizonte, Minas Gerais, Brazil, 2 Pesquisa Clínica e Políticas Públicas em Doenças Infecciosas e Parasitárias, Instituto René Rachou—Oswaldo Cruz Foundation, Belo Horizonte, Minas Gerais, Brazil

* karine.lopes@aluno.fiocruz.br (KFL); edward.oliveira@fiocruz.br (EO)

## Abstract

### Background

Visceral leishmaniasis (VL) is an infectious parasitic disease caused by the species *Leishmania (Leishmania) infantum* in the Mediterranean Basin, the Middle East, Central Asia, South America, and Central America, and *Leishmania (Leishmania) donovani* in Asia and Africa. VL represents the most severe and systemic form of the disease and is fatal if left untreated. Vaccines based on chimeric or multiepitope antigens hold significant potential to induce a highly effective and long-lasting immune response against infections by these parasites. This review systematically compiles data on the efficacy and protective capabilities of chimeric and multiepitope antigens, while also identifying potential immunogenic targets for vaccine development.

### Methodology

A systematic search was conducted by independent reviewers across four databases to assess the efficacy of vaccines based on chimeric or multiepitope antigens against VL. The review included original studies that reported parasite load or positivity rates in animals immunized with these vaccines and subsequently challenged or exposed to *L. infantum* infection in preclinical and clinical studies. Key information was extracted, tabulated, and analyzed, with the risk of bias being assessed using the SYRCLE Risk Tool.

### Principal findings

A total of 22 studies were selected, with only one being a randomized clinical trial. Most of the studies were conducted with mice, followed by dogs and hamsters. The reduction in parasite load varied from 14% to 99.6% and from 1.7 to 9.0 log orders. Limiting dilution was the most used method for assessing parasite load, followed by quantitative real-time polymerase chain reaction (qPCR). Most domains had an uncertain risk of bias due to insufficient information described.

**Data Availability Statement:** All relevant data are within the manuscript and its Supporting Information files.

**Funding:** KFL reiceved PhD scholarship from the Coordination for the Improvement of Higher Education Personnel-CAPES-Brazil (Grant: 88887.481786-2020-00), SMFM and EO were supported by researcher fellows from the Conselho Nacional de Desenvolvimento Científico e Tecnológico- CNPq (Grants: 309994-2023-3 and 301555/2022-2). The funders had no role in study design, data collection and analysis, decision to publish, or preparation of the manuscript.

**Competing interests:** The authors have declared that no competing interests exist.

## Conclusions

Vaccine formulations containing various chimeric or multiepitope antigens have been developed and evaluated in different preclinical trials, with only one advancing to clinical trials and commercialization. However, the findings of this review highlight the promising potential of chimeric and multiepitope antigens as vaccine candidates against VL. The evidence presented could play a crucial role in guiding the rational development of new studies focused on using these antigens for vaccination against VL.

### Author summary

Visceral leishmaniasis (VL) is primarily caused by *Leishmania (Leishmania) infantum* and *Leishmania (Leishmania) donovani*. Vaccines based on chimeric or multiepitope antigens have demonstrated significant potential for eliciting effective and long-lasting immune responses against these parasites. This review systematically evaluates the efficacy of these antigens and identifies potential immunogenic targets for vaccine development. A comprehensive search across four databases identified 22 studies, including one randomized clinical trial. Most research was conducted with mice, with additional studies with dogs and hamsters. The reduction in parasite load ranged from 14% to 99.6% and from 1.7 to 9.0 log orders. Limiting dilution was the most used method for assessing parasitic load, followed by quantitative real-time polymerase chain reaction (qPCR), imprint, biopsy culture, and smear. These findings underscore the promising potential of chimeric and multiepitope antigens as vaccine candidates for VL and provide valuable insights for guiding future vaccine development.

## Introduction

Leishmaniasis is a group of parasitic diseases caused by different species of the protozoan genus *Leishmania* (Kinetoplastida; Trypanosomatidae) and represents one of the major public health concerns in developing countries, according to the World Health Organization [1]. The disease manifests in two main clinical forms, namely cutaneous (CL) or visceral (VL) leishmaniasis, with the latter being the most severe, systemic form and lethal, if not treated. VL can be caused by *Leishmania (Leishmania) donovani*, which is endemic in Asia, Africa, and the Indian subcontinent, where transmission is anthroponotic. It can also be caused by *Leishmania (Leishmania) infantum*, which occurs in Europe and the Americas, extending from south-central Texas to Central and South America. *Leishmania infantum* is characterized by zoonotic transmission, with the domestic dog being the main reservoir of the parasite [2,3]. An estimated 50,000 to 90,000 new cases of VL occur worldwide each year. In 2022, around 85% of VL cases were reported in seven countries, including Brazil [4].

Control strategies for zoonotic VL involve the diagnosis and treatment of human cases; euthanasia of seropositive dogs, their main reservoir; control of the parasite vector; and health education. However, these strategies often fail to yield satisfactory results, leading to an annual increase in both human and canine cases [4]. Given this scenario, the development of new VL control strategies is extremely important. Among these, vaccines stand out as one of the most cost-effective and efficient methods for controlling and preventing the disease [5,6].

Vaccines based on chimeric or multiepitope antigens demonstrate significant potential in generating a highly effective and long-lasting immune response against *Leishmania* spp. [7,8]. Compared to recombinant protein vaccines, these formulations, by using multiple epitopes, increase the probability of interaction with several MHC haplotypes, optimizing immune recognition. Furthermore, this approach offers a safer and more specific alternative, overcoming some of the complexities and risks associated with formulations using live or attenuated parasites [9,10]. Although various preclinical studies have investigated the use of chimeric or multiepitope antigens in vaccine formulations against VL [11–13], their efficacy is influenced by several variables, including the choice of animal model, the proteins used in their construction, and the type of adjuvant. This review systematically compiles data on the effectiveness and protective efficacy of chimeric and multiepitope antigens, while also identifying potential immunogenic targets for vaccine development.

## Methods

### Protocol and registration

Before study initiation, a search was conducted in the International Prospective Register of Systematic Reviews to identify any equivalent studies that had been completed or were ongoing. Since no similar study protocol was identified, the current one was subsequently registered (PROSPERO: CRD42023449370). This systematic review was conducted following the methodological principles of the Cochrane Handbook [14] and adhered to the Preferred Reporting Items for Systematic Reviews and Meta-Analyses (PRISMA) guidelines.

### Eligibility criteria

The guiding question of this study was: What is the efficacy of vaccines based on chimeric or multiepitope antigens for protection against visceral leishmaniasis? In this study, chimeric antigens were defined as molecules formed by the fusion of fragments from various distinct antigens, while multiepitope antigens were defined as those designed through the strategic selection of epitopes derived from different antigens. The PICO framework (Population, Intervention, Comparator, Outcome) was employed in the selection process, defined as follows: (P) patients or animals at risk of developing VL caused by *L. infantum*, or in preclinical studies, were considered animals susceptible to *L. infantum* infection; (I) vaccine based on chimeric antigens or multiepitope antigens; (C) placebo or control groups immunized with adjuvant or saline; and (O) parasite load or positivity rate in studies involving natural infection.

This review included studies that reported parasite load or positivity rate in animals immunized with vaccines based in chimeric or multiepitope antigens (synthetic or recombinant antigens) and challenged or exposed to *L. infantum* infection in preclinical and clinical settings. The exclusion criteria were as follows: I) non-original studies, including literature reviews, editorials, brief communications, and case reports; II) evaluations of vaccine formulations against other *Leishmania* spp.; III) evaluations of immune response or other outcomes without reporting parasite load or positivity rate; IV) in vitro studies; V) studies without a control group; VI) vaccines used as immunotherapy; and VII) publications in languages other than Portuguese, English, or Spanish.

### Search strategy

The search was conducted across four electronic databases, namely MEDLINE (PubMed), Embase, Cochrane Central Registry of Controlled Trials (CENTRAL), and the Virtual Health Library (VHL). For each database, search strategies combining keywords related to "visceral

leishmaniasis" and "vaccines" were combined. Details about the strategy used in each database are available in S1 File. Articles published up to July 27, 2023, were considered, with no restrictions on publication date.

### Selection process

For each database, all retrieved articles were imported to Mendeley Reference Management [15] to identify and remove duplicates. The records were then transferred to Rayyan software [16] for title and abstract screening, which was conducted independently by two reviewers (KLF and MLF) according to inclusion and exclusion criteria, with disagreements being resolved by consensus. The selected studies were read in full by two researchers to confirm eligibility or exclude them considering exclusion criteria. Data were also extracted by two independent researchers (KLF and MLF) with disagreements being previously resolved by a third (EO).

### Data extraction

Relevant data were extracted from each selected article including the first author and year of publication, country of study, animal model, immunogen and adjuvant used, route of administration, parasite and strain, and method of parasite load evaluation. Regarding the construction of chimeric or multiepitope antigens, the original antigens were identified, along with the epitope selection method, alleles used for epitope prediction, the use of spacers and the antigen production method. The results were analyzed according to the parasite load assessment method, considering factors that could influence the outcomes, such as immunogen and adjuvant concentrations, number of doses and intervals between them, parasite concentration and infection site, period between last immunization and experimental challenge or exposure to *L. infantum* infection, and period between challenge and evaluation of the result.

Some considerations were made in summarizing the results. For studies that evaluated a vaccine's efficacy at different points [17], the results were presented from the final evaluation. For studies reporting results as the number of positive animals [18,19], the positivity rate was calculated using the following formula: number of positives/total number in the group × 100.

### Risk of bias

Risk of bias was assessed using the SYRCLE risk tool, which is specifically designed for preclinical studies. Two researchers independently performed all assessments (KFL and MLF), and any discrepancies were resolved by consensus.

## Results

### Literature search

An initial search identified 3,091 articles in the databases. Screening and selection based on eligibility criteria resulted in 343 studies for potential inclusion. To avoid the exclusion of potentially eligible records, the exclusion of studies based on the type of intervention was conducted only during the full-text review, which resulted in 22 studies that met the inclusion criteria. The PRISMA flow diagram summarizes the study selection process and reasons for exclusion (Fig 1).

### Descriptive analysis of included studies

The main characteristics of the 22 included studies are presented in Table 1. They were published in the period of 2003 to 2023. Ten studies were conducted in Brazil [12,20–28], five in

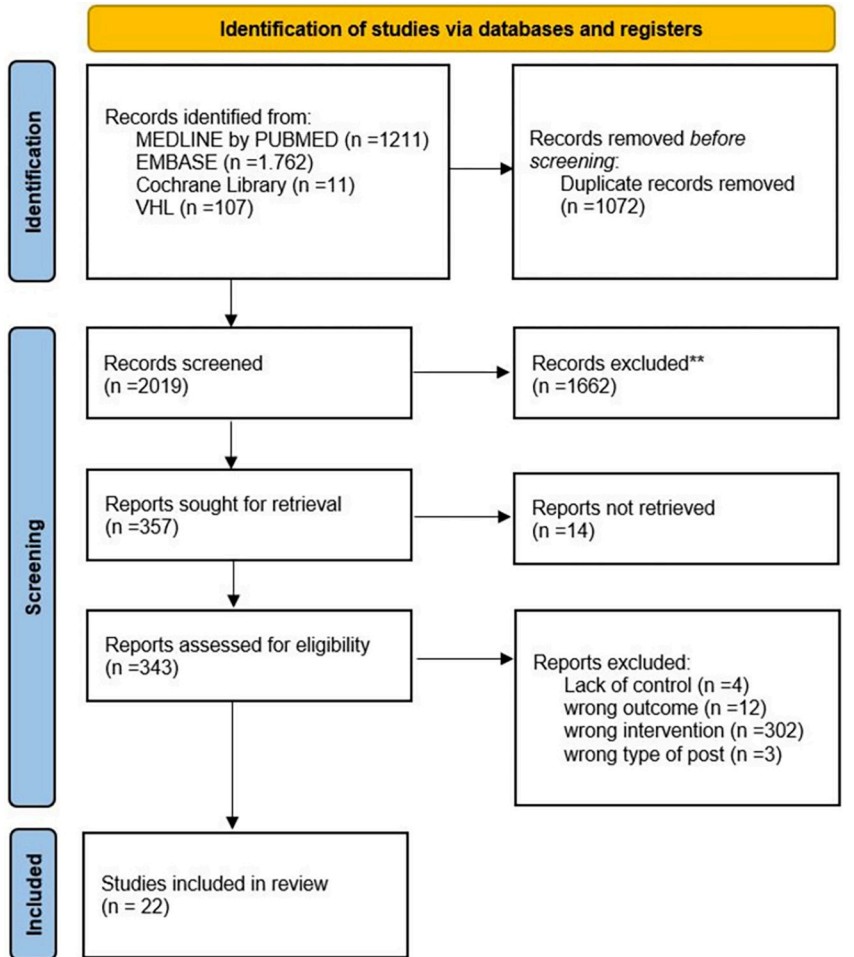

**Fig 1. PRISMA flow diagram of the study selection process.**

Spain [17–19,29,30], three in the USA [31–33], two in Greece [34,35], one in Iran [36] and one in Japan [37]. The main animal model used was the mouse (*Mus musculus*), specifically BALB/ c (15 out of 22 studies), followed by C57BL (4 out of 22 studies) and B6.Cg-Tg (1 out of 22 studies). Additionally, dogs were evaluated in three studies and hamsters (*Mesocricetus auratus*) in only two.

Most of the included studies were preclinical, with only one randomized clinical trial evaluating natural *Leishmania* infection in dogs previously immunized with a multiepitope vaccine [19]. In total, 19 different vaccine formulations were evaluated, mainly using as adjuvant saponin [12,20–23,30], monophosphoryl lipid A [31,32,34,37], or both [24–27]. In preclinical studies, the subcutaneous route was the most used for vaccine evaluation, whether in the paw, base of the tail or back; intraperitoneal and intramuscular routes were less utilized in the included studies.

Challenge with *L. infantum* occurred predominantly intravenously [17,23,24,27,29– 32,34,37], followed by subcutaneously [12,20–22,26,28]. Alternative routes were also used, such as intraperitoneal [25,36], intracardiac [31] and retro-orbital sinus [33], with one study not specifying the route of parasite inoculation [18]. Regarding ethical considerations, only two studies [29,31] included in the review did not explicitly address this aspect for all the different animals models involved in the experiments.

Table 1. The main characteristics of included studies.

| Author | Country of study | Animal model (type; age; sex) | Chimeric or multiepitope vaccine * | Adjuvant | Vaccine administration | Challenge: *Leishmania* species and infection sites | Outcome assessment method |
|---|---|---|---|---|---|---|---|
| Molano et al. 2003 [17] | Spain | BALB/c; 6-week-old; Beagles; 8 to 9 months old | Protein Q | BCG | IP | *L. infantum* (M/CAN/ES/96/BCN), IV tail vein | Smear, impression and culture of the spleen |
| Parody et al. 2004 [29] | Spain | BALB/c; 4 to 6-week-old | Protein Q | CpG-ODN | IP or in the paw | *L. infantum*, IV | Limiting dilution of the spleen and liver |
| Coler et al. 2007 [31] | USA | C57BL/6; 6 to 8-week-old; female Hamster LVG; 30–33 days old | Leish-111f | Monophosphoryl lipid A-stable emulsion | SC in the paw and at the base of the tail | *L. infantum* (MHOM/BR/82/BA-2), IV and intracardiac | Limiting dilution of the spleen and liver |
| Bertholet et al. 2009 [32] | USA | C57BL/6; 5 to 7-week-old; female | Leish-110 | Monophosphoryl lipid A-stable emulsion | SC | *L. infantum*, IV tail vein | Limiting dilution of the spleen |
| Carcelén et al. 2009 [18] | Spain | Beagles; 1 to 2 years old; both sexes | Protein Q | No adjuvant | SC on the left flank | *L. infantum* (M/CAN/ES/96/BCN 150, zimodeme MON-1) | DNA polymerase chain reaction of the skin; biopsy culture and imprint of the spleen and lymph node |
| Goto et al. 2011 [37] | Japan | C57BL/6; 6 to 8-week-old; female | KSAC | Monophosphoryl lipid A-stable emulsion | SC at the base of the tail | *L. infantum* (MHOM/BR/82/BA-2), IV tail vein | Limiting dilution of the spleen and liver |
| Duthie et al. 2017 [33] | USA | C57BL; 6 to 8-week-old; female BALB/c | LEISH-F3 | GLA-SE | SC at the base of the tail | *L. infantum* (MHOM/BR/82/BA-2), retro-orbital sinus | DNA polymerase chain reaction of the liver |
| Athanasiou et al. 2017 [34] | Greece | B6.Cg-Tg (HLA-A/H2-D); 6 to 8-week-old; female | Mix B | Monophosphoryl lipid A | SC | *L.infantum* (MHOM/GR/2001/GH8), IV | Limiting dilution of the spleen and liver |
| Martins et al. 2017 [12] | Brazil | BALB/c; 8-week-old; female | Recombinant chimeric protein (RCP) | Saponin | SC in the left hind paw | *L. infantum* (MHOM/BR/1970/BH46), SC right hind paw | Limiting dilution of the spleen, liver, bone marrow, and draining lymph nodes; DNA polymerase chain reaction of the spleen |
| Cotrina et al. 2018 [19] | Spain | Dogs | Protein Q (LetiFend) | No adjuvant | SC intrascapular | No challenge | DNA polymerase chain reaction and smear of the of bone marrow and lymph nodes |
| Dias et al. 2018 [20] | Brazil | BALB/c; 6 to 8-week-old; female | Chimera | Saponin | SC in the left hind paw | *L. infantum* (MHOM/BR/1970/BH46), SC right hind paw | Limiting dilution of the spleen, liver, bone marrow, and draining lymph nodes; DNA polymerase chain reaction of the spleen |
| Lage et al. 2020 [21] | Brazil | BALB/c; 8-week-old; female | Chimera T | Saponin | SC in the left hind paw | *L. infantum* (MHOM/BR/1970/BH46), SC right hind paw | Limiting dilution of the spleen, liver, bone marrow, and draining lymph nodes; DNA polymerase chain reaction of the spleen |
| Agallou et al. 2020 [35] | Greece | BALB/c; 6 to 8-week-old; female | LiChimera | Addavax | IM | *L. infantum* (MHOM/GR/2001/GH8), IV tail vein | Limiting dilution of the spleen and liver |
| Vakili et al. 2020 [36] | Iran | BALB/c; 6 to 8-week-old; female | Vaccine | Freund | SC | *L. infantum* (MCAN/IR/07/Moheb-gh), IP | Imprint of the spleen |
| Brito et al. 2020 [23] | Brazil | BALB/c; 6 to 8-week-old; female | Chimera A and Chimera B | Saponin | SC on the back | *L. infantum* (MCAN/BR/2008/OP46), IV | DNA polymerase chain reaction of the spleen |

(*Continued*)

**Table 1.** (Continued)

| Author | Country of study | Animal model (type; age; sex) | Chimeric or multiepitope vaccine * | Adjuvant | Vaccine administration | Challenge: *Leishmania* species and infection sites | Outcome assessment method |
|---|---|---|---|---|---|---|---|
| Martínez-Rodrigo et al. 2020 [30] | Spain | BALB/c; 6 to 8-week-old; female | HisDTC e AK | Saponin | SC in the paw | *L. infantum* (MCAN/ES/96/BCN150 zymodeme MON-1), IV | Limiting dilution of the spleen and liver |
| Lage et al. 2020 [22] | Brazil | BALB/c; 8-week-old; female | Chimera T/Liposoma | Saponin | SC in the left hind paw | *L. infantum* (MHOM/BR/1970/BH46), SC right hind paw | Limiting dilution of the spleen, liver, bone marrow, and draining lymph nodes; DNA polymerase chain reaction of the spleen |
| Ostolin et al. 2021 [24] | Brazil | BALB/c; 6 to 8-week-old; female | Chimera A | Saponin and Monophosphoryl lipid A | SC on the back | *L. infantum* (MCAN/BR/2008/OP46), IV | DNA polymerase chain reaction of the spleen |
| Lage et al. 2022 [26] | Brazil | BALB/c; 8-week-old; female | ChimT | Saponin or Monophosphoryl lipid A | SC in the left hind paw | *L. infantum* (MHOM/BR/1970/BH46), SC right hind paw | Limiting dilution of the spleen, liver, bone marrow, and draining lymph nodes; DNA polymerase chain reaction of the spleen |
| Ostolin et al. 2022 [27] | Brazil | BALB/c; 6 to 8-week-old; female | Poly-T Leish | Saponin and Monophosphoryl lipid A | SC | *L. infantum* (MCAN/BR/2008/OP46), IV tail vein | DNA polymerase chain reaction of the spleen and liver |
| Gusmão et al. 2022 [25] | Brazil | Golden hamster; 4 to 6-week-old | Chimera A and Chimera B | Saponin and Monophosphoryl lipid A | SC on the back | *L. infantum* (MCAN/BR/2008/OP46), IP | DNA polymerase chain reaction of the spleen |
| Clímaco et al. 2023 [28] | Brazil | BALB/c; 6 to 8-week-old; female | Chimera | Monophosphoryl lipid A | SC | *L. infantum* (MOM/BR/1970/BH46), SC right hind paw | DNA polymerase chain reaction of the liver |

*SC–subcutaneous; IP–intraperitoneal; IM–intramuscular; IV–intravenous* * For more detail see Table 2.

Table 2 summarizes the main characteristics of the evaluated chimeric and multiepitope antigens, highlighting their constituent proteins, epitope selection methods, use of spacers in the sequence design, and antigen production method. In this context, a total of 17 distinct antigens were constructed, including eleven multiepitope antigens, five chimeric antigens, and one pool of three synthetic peptides designed with selected epitopes from *Leishmania infantum* proteins by prediction analyses.

Only one study did not describe the source proteins used in the construction of the multiepitope antigen [28]. Common proteins across different studies include ribosomal and histones proteins (e.g. Lip2a, Lip2b, P0 and H2A), cysteine peptidases (CPA, CPB, and CPC), and hypothetical proteins (LiHy, LiH1, LiH2, etc.). Most studies described the prediction of MHC class I and II epitopes as their method for epitope selection; however, the approach used was not described in some studies [17–19,29,31–33,37]. The use of spacers was relatively uncommon, being reported in only nine of the 22 included studies. In some cases, antigens previously described in other studies were used and, in these cases, the original studies were consulted. Among these, the amino acid sequence "GGPPG" was the most frequently reported spacer [23–25,27,38]. Most of the included studies used chimeric or multiepitope antigens produced by recombinant DNA technology, employing the bacterium *Escherichia coli* as the expression vector. However, the studies by Athanasiou et al. (2017) [34] and Martínez-Rodrigo et al. (2020) [30] used chemically synthesized antigens, while Cotrina et al. (2018) [19] utilized the

**Table 2. Characteristics of the chimeric or multiepitope antigens of the evaluated vaccine formulations.**

| Author | Source Proteins | Epitope Selection Method | Alleles Used for Epitope Prediction | Uses spacers? If yes, which ones? | Multiepitope vaccine | Chimeric or multiepitope antigen production method |
|---|---|---|---|---|---|---|
| Molano et al. 2003 [17] | Acidic ribosomal proteins Lip2a, Lip2b, P0 and the histone H2A protein from *L. infantum* | NM | NM | No | Protein Q | Expression in *E. coli* |
| Parody et al. 2004 [29] | Acidic ribosomal proteins Lip2a, Lip2b, P0, and the histone H2A protein from *L. infantum* | NM | NM | No | Protein Q | Expression in *E. coli* |
| Coler et al. 2007 [31] | *L. braziliensis* elongation initiation factor (LeIF), *L. major* stress-inducible protein (LmSTI1), and *L. major* thiol-specific antioxidant (TSA) | NM | NM | No | Leish-111f | Expression in *E. coli* |
| Bertholet et al. 2009 [32] | *L. braziliensis* elongation initiation factor (LeIF), *L. major* stress-inducible protein (LmSTI1), and *L. major* thiol-specific antioxidant (TSA) | NM | NM | No | Leish-110 | Expression in *E. coli* |
| Carcelén et al. 2009 [18] | Acidic ribosomal proteins Lip2a, Lip2b, P0, and the histone H2A protein from *L. infantum* | NM | NM | No | Protein Q | Expression in *E. coli* |
| Goto et al. 2011 [37] | *L. infantum* kinetoplastid membrane protein 11 (XM_001468995.1), *L. infantum* sterol 24-c-methyltransferase (XM_001469795.1), *L. donovani* A2 (S69693.1), and *L. infantum* cysteine proteinase B (AJ420286.1) | NM | NM | No | KSAC | Expression in *E. coli* |
| Duthie et al. 2017 [33] | *L. infantum* nucleoside hydrolase (NH), and *L. donovani* sterol-24-c-methyltransferase (SMT) | NM | NM | No | LEISH-F3 | Expression in *E. coli* |
| Athanasiou et al. 2017 [34] | Cysteine peptidase A (CPA), histone H1, and e kinetoplastid membrane protein 11 (KMP11) from *L. infantum* | Prediction of MHC-I and MHC-II epitopes | HLA-A2 (A*0201), HLA-A3 (A*03), HLA-A24 (A*2402), HLA-DRB1, HLA-DPA1, and HLA-DQA1 from human; H2-Db, and H2-Kb from mouse | No | Mix B | Chemical synthesis |
| Martins et al. 2017 [12] | Hypothetical amastigote-specific protein 1 (XP_001468941.1), LiHyp6 (XP_001568689.1), IgE-dependent histamine-releasing factor (HRF) (CAJ05086.1), and hypothetical protein *Leishmania* LiHyV (XP_888524.1) from *L. infantum* | Prediction of MHC-I and MHC-II epitopes | HLA-A2, HLA-A3, and HLA-B7 from human; H-2-Kd, H-2-Ld, H-2-Dd, I-Ad; I-Ed from mouse | No | Recombinant chimeric protein (RCP) | Expression in *E. coli* |
| Cotrina et al. 2018 [19] | Acidic ribosomal proteins Lip2a, Lip2b, P0 and the histone H2A protein from *L. infantum* | NM | NM | No | Protein Q (LetiFend) | Commercial vaccine |
| Dias et al. 2018 [20] | Hypothetical protein S (XP_001467126.1), small glutamine-rich tetratricopeptide repeat-containing (XP_001467120 .1), and prohibitin (XP_001468827.1) from *L. infantum* | Prediction of MHC-I and MHC-II epitopes | HLA-A2, HLA-A3, and HLA-B7 from human; H-2-Kd, H-2-Ld, H-2-Dd, I-Ad, and I-Ed from mouse | No | Chimera | Expression in *E. coli* |

*(Continued)*

**Table 2.** (Continued)

| Author | Source Proteins | Epitope Selection Method | Alleles Used for Epitope Prediction | Uses spacers? If yes, which ones? | Multiepitope vaccine | Chimeric or multiepitope antigen production method |
|---|---|---|---|---|---|---|
| Lage et al. 2020 [21] | Prohibitin (XP_001468827.1), Eukaryotic Initiation Factor 5a (XP_001466105.1), *Leishmania* hypothetical protein 1 (XP_001468941.1), and *Leishmania* hypothetical protein 2 (XP_001462854.1) from *L. infantum* | Prediction of MHC-I and MHC-II epitopes | A2, A3, A24, B7 from human; H-2Db, H-2Dd, H-2Kb, H-2Kd, H-2Kk; H-2Ld; H-2IAb, H-2IAd, H-2Ias, H-2IEd; H-2IEb from mouse | Yes, amino acid sequence GG | Chimera T | Expression in *E. coli* |
| Agallou et al. 2020 [35] | Cyclophilin 2 (XP_001463094), cyclophilin 40 (XP_001469283), enolase (XP_001468063), dihydrolipoamide dehydrogenase (XP_001468025), mitochondrial chaperonin HSP60 (XP_001467869), and a hypothetical protein (XP_001463461) from *L. infantum* | Prediction of MHC-I and MHC-II epitopes | H2-Dd, H2-Kd, H2-Ld, H2-IAd, and H2-IEd from mouse | Yes, amino acid sequence AAY, GGPPG, and EAAAK | LiChimera | Expression in *E. coli* |
| Vakili et al. 2020 [36] | Histone H1, sterol 24-c-methyltransferase (SMT), and *Leishmania*-specific antigenic protein (LSAP) from *L. infantum*, and *Leishmania*-specific hypothetical protein (LiHy) from *L. major* | Prediction of MHC-I and MHC-II epitopes and IFN-Y inducers | H-2Db, H-2Dd, H-2Kb, H-2Kd, H-2Kk, H-2Ld, H-2IAb, H-2IAd, H-2IAs, H-2IEd, and H-2IEb from mouse | Yes, amino acid sequence AAYKK, GSGSGS, and EAAAK | NI | Expression in *E. coli* |
| Brito et al. 2020 [23] | Chimera A: Histone protein (H2A), acid ribosomal protein P2 (LiP2a), acid ribosomal protein P0 (LiP0), *Leishmania* homologue of activated C kinase (LACK), and Cysteine peptidase C (CPC) from *L. infantum* Chimera B: Cysteine peptidase A (CPA), cysteine peptidase B (CPB), surface antigenic protein (PSA-50S), and amastigote protein A2 (A2) from *L. infantum* | Prediction of MHC-I and MHC-II epitopes | HLA-B62, HLA-A*01, HLA-A*02, HLA-A*03, HLA-A*24, HLA-A*26, HLA-B*27, HLA-B*39, HLA-B*44, HLA-B*58, HLA-B*62, HLA-DRB1*0101, HLA-DRB1*0401, HLA-DRB1*0404, HLA-DRB1*0701, HLA-DRB1*0901, HLA-DRB1*1302, HLA-DRB1*1501, HLA-DRB4*0101, and HLA-DRB5*0101 from human; H2-Dd, H2-Dk, and H2-Kd from mouse | Yes, amino acid sequence GGPPG | Chimera A and Chimera B | Expression in *E. coli* |
| Martínez-Rodrigo et al. 2020 [30] | Histone H2A, histone H2B, Histone H3, histone H4, A2 protein, and kinetoplastid membrane protein 11 from *L. infantum* | Prediction of MHC-I and MHC-II epitopes | HLA-A, HLA-B, HLA-C, HLA-E, HLA-DR, HLA-DP, and HLA-DQ from human; H2 from mouse | No | HisDTC and AK | Chemical synthesis |
| Lage et al. 2020 [22] | Prohibitin (XP_001468827.1), Eukaryotic Initiation Factor 5a (XP_001466105.1), *Leishmania* hypothetical protein 1 (XP_001468941.1), and *Leishmania* hypothetical protein 2 (XP_001462854.1) from *L. infantum* | Prediction of MHC-I and MHC-II epitopes | HLA-A2, HLA-A3, HLA-A24, HLA-B7, and HLA-DR from human; H-2Db, H-2Dd, H-2Kb, H-2Kd, H-2Kk, H-2Ld, H-2IAb, H-2IAd, H-2Ias, H-2IEd, and H-2IEb from mouse | Yes, amino acid sequence GG | Chimera T/ Liposome | Expression in *E. coli* |

*(Continued)*

**Table 2.** (Continued)

| Author | Source Proteins | Epitope Selection Method | Alleles Used for Epitope Prediction | Uses spacers? If yes, which ones? | Multiepitope vaccine | Chimeric or multiepitope antigen production method |
|---|---|---|---|---|---|---|
| Ostolin et al. 2021 [24] | Histone protein (H2A), acid ribosomal protein P2 (LiP2a), acid ribosomal protein P0 (LiP0), *Leishmania* homologue of activated C kinase (LACK), and Cysteine peptidase C (CPC) from *L. infantum* | Prediction of MHC-I and MHC-II epitopes | HLA-B62, HLA-A*01, HLA-A*02, HLA-A*03, HLA-A*24, HLA-A*26, HLA-B*27, HLA-B*39, HLA-B*44, HLA-B*58, HLA-B*62, HLA-DRB1*0101, HLA-DRB1*0401, HLA-DRB1*0404, HLA-DRB1*0701, HLA-DRB1*0901, HLA-DRB1*1302, HLA-DRB1*1501, HLA-DRB4*0101, and HLA-DRB5*0101 from human; H2-Dd, H2-Dk, and H2-Kd from mouse | Yes, amino acid sequence GGPPG | Chimera A | Expression in *E. coli* |
| Lage et al. 2022 [26] | *Leishmania* hypothetical protein 1 (XP_001468941.1), *Leishmania* hypothetical protein V (XP_001462854.1), *Leishmania* hypothetical protein C (XP_001470432.1), and *Leishmania* hypothetical protein G (XP_001467126.1) from *L. infantum* | Prediction of MHC-I and MHC-II epitopes | HLA-A2, HLA-A3, HLA-A24, HLA-B7, and HLA-DR from human; H-2Db, H-2Dd, H-2Kb, H-2Kd, H-2Kk, H-2Ld, H-2IAb, H-2IAd, H-2Ias, H-2IEd, and H-2IEb from mouse | Yes, amino acid sequence GG, and KK | ChimT | Expression in *E. coli* |
| Ostolin et al. 2022 [27] | Cysteine peptidase A (CPA), cysteine peptidase B (CPB), surface antigenic protein (PSA-50S), and amastigote protein A2 (A2) from *L. infantum* | Prediction of MHC-I and MHC-II epitopes | HLA-B62, HLA-A*01, HLA-A*02, HLA-A*03, HLA-A*24, HLA-A*26, HLA-B*27, HLA-B*39, HLA-B*44, HLA-B*58, HLA-B*62, HLA-DRB1*0101, HLA-DRB1*0401, HLA-DRB1*0404, HLA-DRB1*0701, HLA-DRB1*0901, HLA-DRB1*1302, HLA-DRB1*1501, HLA-DRB4*0101, and HLA-DRB5*0101 from human; H2-Dd, H2-Dk, and H2-Kd from mouse | Yes, amino acid sequence GGPPG | Poly-T Leish | Expression in *E. coli* |
| Gusmão et al. 2022 [25] | Chimera A: Histone protein (H2A), acid ribosomal protein P2 (LiP2a), acid ribosomal protein P0 (LiP0), *Leishmania* homologue of activated C kinase (LACK), and Cysteine peptidase C (CPC) from *L. infantum* Chimera B: Cysteine peptidase A (CPA), cysteine peptidase B (CPB), surface antigenic protein (PSA-50S), and amastigote protein A2 (A2) from *L. infantum* | Prediction of MHC-I and MHC-II epitopes | | Yes, amino acid sequence GGPPG | Chimera A and Chimera B | Expression in *E. coli* |
| Clímaco et al. 2023 [28] | NI | Prediction of MHC-I and MHC-II epitopes | HLA-A2, HLA-A3, and HLA-B7 from human | No | Chimera | Expression in *E. coli* |

*NI–Not informed*: the original article did not include any information about the specific item; *NM–No made*: the analysis was not conducted in the studies

manufactured LetiFend vaccine that contains Protein Q as the active pharmaceutical ingredient.

Here, parasite load was considered the main outcome of interest to understand the vaccine efficacy of different formulations. However, there was no consensus on the evaluation methodology, the evaluated organs, or the best way to present the results. Therefore, Tables 3 to 5 shown the outcomes according to the methodology used to evaluate parasite load.

Limiting dilution was the most frequently used methodology (12/22) to assess parasite load in the spleen, liver, bone marrow, and lymph nodes (Table 3). In these studies, parasite load was mainly evaluated after three doses of vaccine with 14 days between them. For the challenge, parasite dose in the infection ranged from $5 \times 10^5$ to $1 \times 10^7$. These studies compared the parasite load of organs of animals immunized with chimeric or multiepitope antigens with the load of animals immunized with saline, adjuvant, or non-immunized. Parasite load reduction was demonstrated through a percentage, ranging from 14% [35] to 99.6% [31], or through log orders, ranging from 1.7 [26] to 9.0 [21]. However, some studies only presented qualitative descriptions of the results without quantifying the reduction in parasite load [12,20,30,32].

Several studies independently evaluated formulations containing a multiepitope antigen and those containing the multiepitope antigen associated with adjuvants. The addition of adjuvants is generally associated with a greater reduction in parasite load [12,29,35]. Animals immunized with Protein Q, Leish-111f, or Chimera T, experienced a reduction in parasite load of more than 90%, particularly in the spleen and liver, compared to animals immunized with saline [17,22,31]. Notably, Chimera T was the only immunogen that achieved this high reduction in parasite load without the need for an adjuvant [22].

The studies that assessed parasite load using qPCR are described in Table 4. Among these, only one used dog as an animal model, immunizing them with one or two doses of Protein Q, and observed variable positivity, ranging from 14.4% to 57.1%, in the skin detected by qPCR after experimental infection. Most of the studies evaluated vaccines in mice or hamsters and used three doses with intervals of 14 or 21 days.

The spleen was the most frequently evaluated organ [12,20,22–26], followed by the liver [28,33], or both organs [27], in addition to the skin [18]. Kinetoplast minicircle DNA (kDNA) was the most employed molecular target for quantifying parasite load, used in nine studies. Only two studies utilized different targets, namely genomic repeat region specific to *Leishmania* species [33] and DNA polymerase [23]. The reduction in parasite load ranged from 41.9% [12] to 96% [25], or was reduced by 2 to 7 times [26].

A limited number of studies employed alternative methodologies to assess parasite load, including imprinting [17,18,36], biopsy culture [17,18] and organ smears [17] (Table 5). Parasite load reduction in murine model ranged from 72% to 81%, while in dogs the antigen protein achieved a protective efficacy of 90% [17]. Furthermore, Carcelén et al. (2009) [18] presented results based on the number of negative dogs, while the effectiveness of the tested formulation was not quantified in Vakili et al. (2020) [36], although a significant reduction was found.

Table 6 presents the data from the only clinical trial included. The LetiFend vaccine, based on Protein Q, was evaluated in 168 dogs compared to a placebo group, with both groups being exposed to a natural infection and receiving a single vaccine dose. The results were analyzed based on animal infection, assessed by qPCR and bone marrow and lymph node smears. The infection positivity rate in the placebo group was 16.1%, while in the vaccinated group it was 7.15%, as detected by qPCR. These results were consistent with the smear findings, where the positivity rate was 5.35% in the vaccinated group and 13.9% in the placebo group [19].

**Table 3. Parasite load assessed by limiting dilution reported by the different included studies.**

| Authors | Animal model | Formulation with chimeric or multiepitope chimeric vaccine | Number of doses (interval between doses) | *L. infantum* concentration and infection site | Days between last immunization and challenge | Days between infection and outcome assessment | Parasite load assessment compared to control group: | | |
|---|---|---|---|---|---|---|---|---|---|
| | | | | | | | Saline | Adjuvant | Non-immunized |
| Parody et al. 2004 [29] | Mouse (BALB/c) | Protein Q (2 μg) | 2 x (15 days) | $1 \times 10^6$ IV | 21 | 28 | ↓ 50% in the spleen and in the liver | NA | NA |
| | Mouse (BALB/c) | Protein Q (2μg) + CPg-ODN (20 μg) | 2 x (15 days) | $1 \times 10^6$ IV | 21 | 28 | ↓ 99% in the spleen and in the liver | NA | NA |
| | Mouse (BALB/c) | Protein Q (2μg) + CPg-ODN (20 μg) | 2 x (15 days) | $1 \times 10^6$ IV | 98 | 28 | ↓ 89% in the spleen | NA | NA |
| Coler et al. 2007 [31] | Mouse (C57BL/6) | Leish-111f (10 μg) + MPL-SE (20 μg) | 3 x (21 days) | $5 \times 10^6$ IV | 30 | 35 | ↓ 91.7% in the liver | NA | NA |
| | Hamster LVG | Leish-111f (10 μg) + MPL-SE (20 μg) | 3 x (21 days) | $5 \times 10^6$ intracardiac | 30 | 30 | ↓ 99.6% in the spleen; NSr in liver | NA | NA |
| Bertholet et al. 2009 [32] | Mouse (C57BL/6) | Leish-110 (10 μg) + MPL-SE (20 μg) | 3 x (14 days) | $5 \times 10^6$ IV tail vein | NI | 28 | SR in the spleen | NA | NA |
| Goto et al. 2011 | Mouse (C57BL/6) | KSAC (10 μg) + MPL-SE (20 μg) | 3 x (21 days) | $5 \times 10^6$ IV tail vein | 21 | 28 | ↓ 66% in the spleen and 93% in the liver | NA | NA |
| Athanasiou et al. 2017 [34] | Mouse B6. Cg-Tg (HLA-A/ H2-D) | Mix B (6 μg) + MPLA (3 μg) | 3 x (14 days) | $2 \times 10^7$ IV | 14 | 30 | NA | NA | ↓ 61.98% in the spleen e 72.91% in the liver |
| | Mouse B6. Cg-Tg (HLA-A/ H2-D) | Mix B (6 μg) + MPLA (3 μg) | 3 x (14 days) | $2 \times 10^7$ IV | 14 | 60 | NA | NA | ↓ 73.64% in the spleen and 64.4% in the liver |
| Martins et al. 2017 [12] | Mouse (BALB/c) | Recombinant chimeric protein (RCP) (25 μg) + saponin (25 μg) | 3 x (14 days) | $1 \times 10^7$ SC right hind paw | 30 | 70 | SR in the spleen, liver, bone marrow and draining lymph nodes | NA | NA |
| | Mouse (BALB/c) | Recombinant chimeric protein (RCP) (25 μg) | 3 x (14 days) | $1 \times 10^7$ SC right hind paw | 30 | 70 | NSr in the spleen, liver, bone marrow and draining lymph nodes | NSr in the spleen, liver, bone marrow and draining lymph nodes | NA |
| Dias et al. 2018 [20] | Mouse (BALB/c) | Chimera (25 μg) + saponin (25 μg) | 3 x (14 days) | $1 \times 10^7$ SC right hind paw | 60 | 60 | SR in the spleen, liver, bone marrow and draining lymph nodes | SR in the spleen, liver, bone marrow and draining lymph nodes | NA |
| | Mouse (BALB/c) | Chimera (15 μg) | 3 x (14 days) | $1 \times 10^7$ SC right hind paw | 60 | 60 | NSr in the spleen, liver, bone marrow and draining lymph nodes | NSr in the spleen, liver, bone marrow and draining lymph nodes | NA |

*(Continued)*

**Table 3.** (Continued)

| Authors | Animal model | Formulation with chimeric or multiepitope chimeric vaccine | Number of doses (interval between doses) | *L. infantum* concentration and infection site | Days between last immunization and challenge | Days between infection and outcome assessment | Parasite load assessment compared to control group: | | |
|---|---|---|---|---|---|---|---|---|---|
| | | | | | | | **Saline** | **Adjuvant** | **Non-immunized** |
| Lage et al. 2020 [21] | Mouse (BALB/c) | Chimera T (15 µg) + saponin (15 µg) | 3 x (14 days) | $1 \times 10^7$ SC right hind paw | 30 | 45 | ↓ ~6.0–9.0 log orders of magnitude in the spleen, liver, bone marrow and draining lymph nodes | ↓ ~6,0–9,0 log orders of magnitude in the spleen, liver, bone marrow and draining lymph nodes | NA |
| Agallou et al. 2020 [35] | Mouse (BALB/c) | LiChimera (10 µg) + Addavax (40 µL) | 2 x (14 days) | $1 \times 10^7$ IV tail vein | 14 | 56 | ↓ 73% in the spleen and 98% in the liver | NA | NA |
| | Mouse (BALB/c) | LiChimera (10 µg) | 2 x (14 days) | $1 \times 10^7$ IV tail vein | 14 | 56 | ↓ 46% in the spleen and 98% in the liver | NA | NA |
| | Mouse (BALB/c) | LiChimera (10 µg) + Addavax (40 µL) | 2 x (14 days) | $1 \times 10^7$ IV tail vein | 14 | 84 | ↓ 83% in the spleen and 88% in the liver | NA | NA |
| | Mouse (BALB/c) | LiChimera (10 µg) | 2 x (14 days) | $1 \times 10^7$ IV tail vein | 14 | 84 | ↓ 14% in the spleen and 37% in the liver | NA | NA |
| Martínez-Rodrigo et al. 2020 [30] | Mouse (BALB/c) | HisDTC (50 µg) + saponin (25 µg) | 3 x (15 days) | $5 \times 10^5$ IV | 30 | 42 | SR in the spleen and in the liver | SR in the spleen and in the liver | NA |
| | Mouse (BALB/c) | AK (50 µg) + saponin (25 µg) | 3 x (15 days) | $5 \times 10^5$ IV | 30 | 42 | SR in the liver; NSr in spleen | SR in the liver; NSr in spleen | NA |
| | Mouse (BALB/c) | HisDTC (25 µg) + AK (25 µg) + saponin | 3 x (15 days) | $5 \times 10^5$ IV | 30 | 42 | SR in the spleen and in the liver | SR in the spleen and in the liver | NA |
| Lage et al. 2020 [22] | Mouse (BALB/c) | Chimera T (20 µg) | 3 x (14 days) | $1 \times 10^6$ SC right hind paw | 30 | 45 | ↓ 98% in the spleen and in the draining lymph nodes, and 94% in the liver and bone marrow | NA | NA |
| | Mouse (BALB/c) | Chimera T (20 µg) + saponin (20 µg) | 3 x (14 days) | $1 \times 10^6$ SC right hind paw | 30 | 45 | ↓ > 99.5% in the spleen, liver, bone marrow and draining lymph nodes | NA | NA |
| | Mouse (BALB/c) | Chimera T (20 µg) /Liposome | 3 x (14 days) | $1 \times 10^6$ SC right hind paw | 30 | 45 | ↓ > 99.5% in spleen, liver, bone marrow and draining lymph nodes | NA | NA |

(*Continued*)

**Table 3.** (Continued)

| Authors | Animal model | Formulation with chimeric or multiepitope chimeric vaccine | Number of doses (interval between doses) | *L. infantum* concentration and infection site | Days between last immunization and challenge | Days between infection and outcome assessment | Parasite load assessment compared to control group: | | |
| --- | --- | --- | --- | --- | --- | --- | --- | --- | --- |
| | | | | | | | Saline | Adjuvant | Non-immunized |
| Lage et al. 2022 [26] | Mouse (BALB/c) | ChimT (20 µg) + saponin (20 µg) | 3 x (14 days) | $1 \times 10^7$ SC right hind paw | 30 | 45 | NA | ↓ 4.0 log orders of magnitude in the spleen, 2.5 in the liver, 1.7 in the bone marrow and 4.7 in the draining lymph nodes | NA |
| | Mouse (BALB/c) | ChimT (20 µg) +MPLA (20 µg) | 3 x (14 days) | $1 \times 10^7$ SC right hind paw | 30 | 45 | NA | ↓ 4.5 log orders of magnitude in the spleen, 3.2 in the liver, 2.5 in the bone marrow and 4.5 in the draining lymph nodes | NA |
| | Mouse (BALB/c) | ChimT (20 µg) | 3 x (14 days) | $1 \times 10^7$ SC right hind paw | 30 | 45 | NSr in the spleen, liver, bone marrow and draining lymph nodes | No Sr in the spleen, liver, bone marrow and draining lymph nodes | NA |

SC–subcutaneous; IV–intravenous; NSr–non-significant results; SR–significant results; NA–not assessed

## Risk of bias assessment

The risk of bias assessment is presented in Fig 2. Except for detection bias, all other domains predominantly indicate an uncertain risk. This is due to insufficient information or inadequate descriptions in many of the evaluated studies. Conversely, most studies showed a low risk of bias in the detection bias domain.

## Discussion

The development of a safe, immunogenic, and effective vaccine is crucial for controlling VL. Vaccines based on chimeric or multiepitope antigens represent a promising approach, combining multiple epitopes into a single sequence to enhance the immune response and provide broader protection against the antigenic variability of the parasite [9,10]. Although some review articles on chimeric or multiepitope vaccines against VL have been published [39–42], a systematic evaluation of the tested vaccine candidates has not yet been conducted. The initial finding of the present review highlights the efforts made in this field but reveals a significant gap between preclinical and clinical studies. Out of 22 articles retrieved through the search strategy, only one was identified as a randomized clinical trial [19]. This finding also highlights the absence of a strategic plan for developing vaccines against VL. A recurring pattern emerged among the 22 selected studies, characterized by the repeated use of the same antigens across various approaches. For instance, Protein Q was investigated in multiple studies, both alone and in combination with various adjuvants [17,29]. It was also explored in research assessing different immunization protocols [18]. Protein Q was engineered by fusion of five intracellular antigenic fragments of acidic ribosomal proteins (LiP2a, LiP2b and LiP0) and histone A2 protein. The antigenicity of Protein Q was first evaluated in an enzyme-linked immunosorbent

**Table 4. Parasite load assessed by DNA polymerase chain reaction (qPCR) reported by the different included studies.**

| Authors | Animal model | Formulation with chimeric or multiepitope vaccine | Number of doses (interval between doses) | *L. infantum* concentration and infection site | Days between last immunization and challenge | Days between infection and outcome assessment | qPCR target for parasitic load | Parasite load assessment compared to control group: | | |
|---|---|---|---|---|---|---|---|---|---|---|
| | | | | | | | | Saline | Adjuvant | Other controls |
| Carcelén et al. 2009 [18] | Dog (Beagle) | Protein Q (100 μg) | Single dose | $5x10^5$ | 60 | 330 | Kinetoplast minicircle DNA | NA | NA | Positivity rate = 14.4% |
| | Dog (Beagle) | Protein Q (100 μg) | 2 x (21 days) | $5x10^5$ | 39 | 330 | Kinetoplast minicircle DNA | NA | NA | Positivity rate = 57.1% |
| Duthie et al. 2017 [33] | Mouse (C57BL/6) | LEISH-F3 (5 μg) + GLA-SE (5 μg) | 3 x (21 days) | $1x10^6$ retro-orbital sinus | 30 | 28 | genomic repeat region specific to *Leishmania* species | Sr in the liver | NA | NA |
| | Mouse (BALB/c) | LEISH-F3 (5 μg) + GLA-SE (5 μg) | 3 x (21 days) | $1x10^6$ retro-orbital sinus | 30 | 28 | genomic repeat region specific to *Leishmania* species | Sr in the liver | NA | NA |
| Martins et al. 2017 [12] | Mouse (BALB/c) | RCP (25 μg) + saponin (25 μg) | 3 x (14 days) | $1x10^7$SC right hind paw | 30 | 70 | Kinetoplast minicircle DNA | ↓ 49.8% in the spleen | ↓ 41.9% in the spleen | NA |
| | Mouse (BALB/c) | RCP (25 μg) | 3 x (14 days) | $1x10^7$SC right hind paw | 30 | 70 | Kinetoplast minicircle DNA | NSr in the spleen | NSr in the spleen | NA |
| Dias et al. 2018 [20] | Mouse (BALB/c) | RCP (25 μg) + Saponin (25 μg) | 3 x (14 days) | $1x10^7$ SC right hind paw | 60 | 60 | Kinetoplast minicircle DNA | ↓ 91% in the spleen | ↓ 88% in the spleen | NA |
| Brito et al. 2020 [20] | Mouse (BALB/c) | Chimera A (10 μg) + saponin (60 μg) | 3 x (15 days) | $1x10^7$ IV | 15 | 30 | DNA polymerase | ↓ 82% in the spleen | ↓ 82% in the spleen | NA |
| | Mouse (BALB/c) | Chimera B (10 μg) + saponin (60 μg) | 3 x (15 days) | $1x10^7$ IV | 15 | 30 | DNA polymerase | ↓ 87% in the spleen | ↓ 87% in the spleen | NA |
| Lage et al. 2020 [22] | Mouse (BALB/c) | Chimera T (20 μg) | 3 x (14 days) | $1x10^6$ SC right hind paw | 30 | 45 | Kinetoplast minicircle DNA | ↓ 45% in the spleen | NA | NA |
| | Mouse (BALB/c) | Chimera T (20 μg) (20 μg) + saponin (20 μg) | 3 x (14 days) | $1x10^6$ SC right hind paw | 30 | 45 | Kinetoplast minicircle DNA | ↓ 65% in the spleen | NA | NA |
| | Mouse (BALB/c) | Chimera T (20 μg)/ Liposome | 3 x (14 days) | $1x10^6$ SC right hind paw | 30 | 45 | Kinetoplast minicircle DNA | ↓ 77% in the spleen | NA | NA |
| Ostolin et al. 2021 [24] | Mouse (BALB/c) | Chimera A (20 μg) + Saponin (30 μg) + MPLA (12,5 μg) | 3 x (14 days) | $1x10^7$ IV | 14 | 42 | Kinetoplast minicircle DNA | ↓ 92% in the spleen | ↓ 92% in the spleen | NA |
| | Mouse (BALB/c) | Chimera A (20 μg) | 3 x (14 days) | $1x10^7$ IV | 14 | 42 | Kinetoplast minicircle DNA | NSr in the spleen | NSr in the spleen | NA |

(*Continued*)

**Table 4.** (Continued)

| Authors | Animal model | Formulation with chimeric or multiepitope vaccine | Number of doses (interval between doses) | *L. infantum* concentration and infection site | Days between last immunization and challenge | Days between infection and outcome assessment | qPCR target for parasitic load | Parasite load assessment compared to control group: | | |
| --- | --- | --- | --- | --- | --- | --- | --- | --- | --- | --- |
| | | | | | | | | Saline | Adjuvant | Other controls |
| Lage et al. 2022 [26] | Mouse (BALB/c) | ChimT (20 μg) + Saponin (20 μg) | 3 x (14 days) | 1x10$^7$SC right hind paw | 30 | 45 | Kinetoplast minicircle DNA | ↓ 4x in the spleen | ↓ 4x in the spleen | NA |
| | Mouse (BALB/c) | ChimT (20 μg) + MPLA (20 μg) | 3 x (14 days) | 1x10$^7$SC right hind paw | 30 | 45 | Kinetoplast minicircle DNA | ↓ 7x in the spleen | ↓ 7x in the spleen | NA |
| | Mouse (BALB/c) | ChimT (20 μg) | 3 x (14 days) | 1x10$^7$SC right hind paw | 30 | 45 | Kinetoplast minicircle DNA | ↓ 2x in the spleen | NA | NA |
| Ostolin et al. 2022 [27] | Mouse (BALB/c) | Poly-T Leish (10 μg) + Saponin (30 μg) + MPLA (12,5 μg) | 3 x (14 days) | 1x10$^7$ IV tail vein | 14 | 45 | Kinetoplast minicircle DNA | ↓ 96% in the spleen and in the liver | ↓ 96% in the spleen and in the liver | NA |
| | Mouse (BALB/c) | Poly-T Leish (10 μg) | 3 x (14 days) | 1x10$^7$ IV tail vein | 14 | 45 | Kinetoplast minicircle DNA | NSr in the spleen and in the liver | NSr in the spleen and in the liver | NA |
| Gusmão et al. 2022 [25] | Hamster | Chimera A (20 μg) + Saponin (50 μg) + MPLA (12,5 μg) | 3 x (15 days) | 2x10$^7$ IP | 21 | 60 | Kinetoplast minicircle DNA | ↓ 87% in the spleen | NA | NA |
| | Hamster | Chimera A (20 μg) | 3 x (15 days) | 2x10$^7$ IP | 21 | 60 | Kinetoplast minicircle DNA | ↓ 76% in the spleen | NA | NA |
| | Hamster | Chimera B (20 μg) + Saponin (50 μg) + MPLA (12,5 μg) | 3 x (15 days) | 2x10$^7$ IP | 21 | 60 | Kinetoplast minicircle DNA | ↓ 92.5% in the spleen | NA | NA |
| | Hamster | Chimera B (20 μg) | 3 x (15 days) | 2x10$^7$ IP | 21 | 60 | Kinetoplast minicircle DNA | ↓ 84% in the spleen | NA | NA |
| Clímaco et al. 2023 [28] | Mouse (BALB/c) | Chimera (10 μg) | 3 x (14 days) | 1x10$^6$ SC right hind paw | 30 | 120 | Kinetoplast minicircle DNA | NA | NSr in the liver | NSr in liver compared to MPLA-associated chimeric protein |
| | Mouse (BALB/c) | Chimera (10 μg) + MPLA (10 μg) | 3 x (14 days) | 1x10$^6$ SC right hind paw | 30 | 120 | Kinetoplast minicircle DNA | NA | ↓ 73% in the liver | ↓ 73% in the liver compared to chimeric protein alone |

*SC–subcutaneous; IV–intravenous; IP–intraperitoneal; NSr–non-significant results; SR–significant results; NA–not assessed*

assay (ELISA) and Western blotting technique, using canine serum samples. Later, Protein Q was evaluated in an ELISA assay and presented sensitivity of 79 to 93% and specificity of 96 to 100% for canine VL diagnosis [43]. These results, associated with those about the capacity of the Lip2a protein to induce proliferation of naïve splenocytes and induce IFN-γ production

**Table 5. Parasite load assessed by other methodologies in the different included studies.**

| Authors | Animal model | Formulation with chimeric or multiepitope chimeric vaccine | Number of doses (interval between doses) | *L. infantum* concentration and infection site | Days between last immunization and challenge | Days between infection and outcome assessment | Outcome assessment method | Parasite load assessment compared to control group: Saline | Adjuvant | Other controls |
|---|---|---|---|---|---|---|---|---|---|---|
| Molano et al. 2003 [17] | Mouse (BALB/c) | Protein Q (2 µg) + BCG (50.000 PFU) | 3 x (14 days) | 1x10$^5$ IV tail vein | 14 | 30 | Smears | ↓ 80% in the spleen | ↓ 72% in the spleen | NA |
| | Mouse (BALB/c) | Protein Q (2 µg) + BCG (50.000 PFU) | 3 x (14 days) | 1x10$^5$ IV tail vein | 14 | 30 | Imprint | ↓ 81% in the spleen | ↓ 73% in the spleen | NA |
| | Dog (Beagle) | Protein Q (4 µg/Kg) + BCG (1.000.000 PFU) | 3x (21 days; 23 days) | 5x10$^5$ IV | 66 | 634 | Biopsy culture | 90% protective efficacy | NA | NA |
| Carcelén et al. 2009 [18] | Dog (Beagle) | Protein Q (100 µg) | single dose | 5x10$^5$ | 60 | 330 | Biopsy culture | 3 dogs with negative spleen culture out of 7 evaluated; 2 dogs with negative lymph node out of 7 evaluated | NA | NA |
| | Dog (Beagle) | Protein Q (100 µg) | single dose | 5x10$^5$ | 60 | 330 | Imprint | 7 dogs with negative spleen culture out of 7 evaluated; 6 dogs with negative lymph node out of 7 evaluated | NA | NA |
| | Dog (Beagle) | Protein Q + Protein Q (100 µg) | 2 x (21 dias) | 5x10$^5$ | 39 | 330 | Biopsy culture | 3 dogs with negative spleen culture out of 7 evaluated; 1 dog with negative lymph node culture out of 7 evaluated | NA | NA |
| | Dog (Beagle) | Protein Q + Protein Q (100 µg) | 2 x (21 days) | 5x10$^5$ | 39 | 330 | Imprint | 5 dogs with negative spleen culture out of 7 evaluated; 6 dogs with negative lymph node culture out of 7 evaluated | NA | NA |
| Vakili et al. 2020 [36] | Mouse (BALB/c) | Vaccine (30 µg) | 2 x (21 days) | 3x10$^6$ IP | 21 | 30 | Imprint | SR in the spleen | SR in the spleen | NSr compared to the peptide Vaccine + adjuvant |

*IV–intravenous; IP–intraperitoneal; SR–significant results; NA–not assessed*

[44] led the research group to test Protein Q associated with the Bacillus of Calmette & Guerin (BCG) in mouse and dogs. The study reported an 81% reduction in parasite load in mice and a 90% efficacy in dogs [17]. Following this, Parody et al. 2004 [29] demonstrated that

**Table 6. Positivity rate of immunized dogs exposed to a natural infection.**

| Author | Animal model | Multiepitope vaccine | Number of doses (interval between doses) | Follow-up days (between the first dose and outcome assessment) | Outcome assessment method | Positivity (%) | |
|---|---|---|---|---|---|---|---|
| | | | | | | Vaccinated | Placebo |
| Cotrina et al. 2018 [19] | Dog | 0.5 mL of LetiFend (Protein Q) | 2 x (365 days) | 730 | qPCR (bone marrow and lymph node) | 7.15 | 16.1 |
| | Dog | 0.5 mL of LetiFend (Protein Q) | 2 x (365 days) | 730 | Smear (bone marrow and lymph node) | 5.35 | 13.9 |

immunizing BALB/c mice with Protein Q plus CpG motifs induced a predominant IgG2a response and provided significant protection against *L. infantum* infection, leading to an 89% reduction in parasite load in the spleen [29]. Subsequently, Protein Q was tested in a double-blind placebo-controlled experiment with dogs, administered as either a single or double dose without adjuvant. Interestingly, the single-dose protocol provided greater protection than the double-dose protocol, with only one out of seven dogs testing positive compared to four out of seven, respectively [18]. The most recent study included in the present review evaluated the effectiveness of Protein Q in a commercial vaccine formulation (LetiFend) in a large-scale, multicenter, randomized, double-blind, placebo-controlled field study [19]. The authors reported a statistically significant reduction in canine VL cases and in the number of parasite-positive dogs in the vaccine group ($p = 0.0564$). In summary, the overall efficacy of Protein Q (LetiFend vaccine) against canine VL was 72% when used for immunizing dogs living in endemic areas. Another notable example is the Leish-111f protein, which is composed of a thiol-specific antioxidant (TSA), *L. major* stress-inducible protein (LmSTI1), and the 26 kDa N-terminal portion of *Leishmania* elongation initiation factor (LeIF), fused in tandem. This antigen was analyzed by Coler et al. (2007) [31], who demonstrated that Leish-111f, when associated with either rIL-12 or 4′-monophosphoryl lipid A plus squalene (MPL–SE), induced significant protection against *L. infantum* infection in mice and hamsters, resulting in a 99.6% reduction in parasite load. However, for production purposes, and due to regulatory concerns, Leish-111f was modified by removing a 6-histidine sequence near the amino-terminal region and by mutating a proteolytic cleavage site, resulting in the Leish-110 protein [32,45]. Notably, Leish-110 combined with MPL-SE also demonstrated a statistically significant reduction in liver parasite load in mice immunized and challenged with *L. infantum*, when compared to both the saline control ($p < 0.01$) and the adjuvant-alone control ($p < 0.05$) [32].

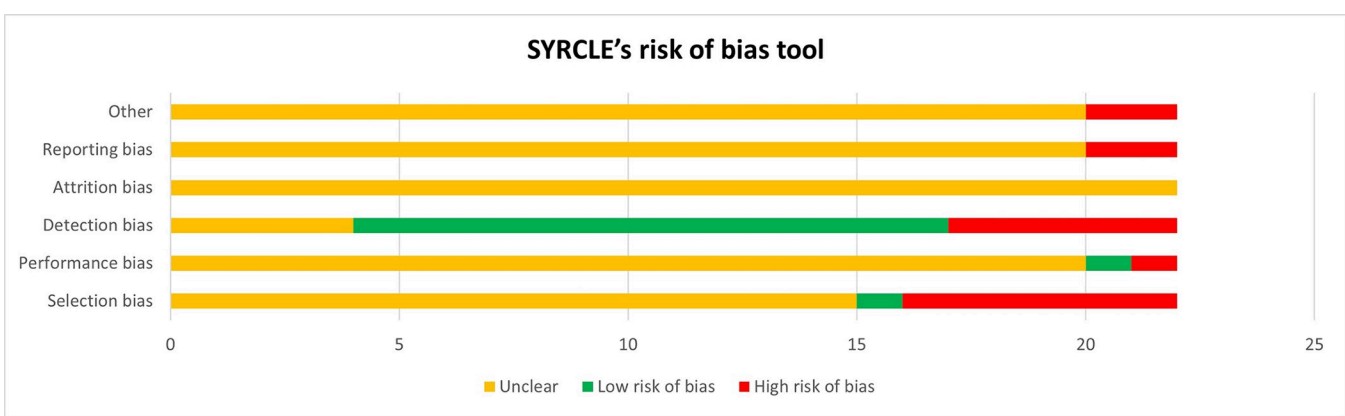

**Fig 2. Risk of bias observed in the included studies.**

Chimera A and B antigens have been extensively evaluated in several studies. Chimeric A antigen is composed of epitopes from histone (H2A) protein, acid ribosomal protein (LiP2a), acid ribosomal protein P0 (LiP0), *Leishmania* homologue of activated C kinase (LACK), and cysteine peptidase C (CPC). In contrast, Chimeric B antigen is composed by epitopes from cysteine peptidase A and B antigens (CPA and CPB), surface antigenic antigen (PSA-50S), and specific amastigote antigen A2 (A2), with GPGPG amino acid sequences intercalated as spacers [23]. Initially, both chimeric antigens were constructed and evaluated in association with the saponin adjuvant. Chimera A resulted in an 82% reduction in parasite load in the spleen, while Chimera B led to an 87% reduction in the same organ [23]. Subsequently, the immune response to different doses of Chimera A was investigated. After determining that the optimal dose was 20 μg, the efficacy of the antigen was evaluated both alone and in combination with the saponin and monophosphoryl lipid A adjuvant system. Although Chimera A alone did not induce a reduction in the parasite load in the spleen, Chimera A associated with the adjuvant system resulted in a 92% reduction [24]. The Chimera B antigen, later renamed Poly-T Leish, was also evaluated with this same adjuvant system and led to a 96% reduction in parasite load in both the spleen and liver [27]. Both chimeric proteins, when tested in a hamster model using the saponin and monophosphoryl lipid A adjuvant system, were able to induce a reduction in parasite load ranging from 76% to 92.5%, corroborating the findings observed in the murine model [25]. In addition, the immunogenicity and vaccine efficacy of the Chimera T antigen, which contains specific T-cell epitopes from *Leishmania* Prohibitin, Eukaryotic Initiation Factor 5a and the hypothetical LiHyp1 and LiHyp2 proteins, have been widely evaluated. Chimera T was evaluated in two studies, one in association with the saponin adjuvant [21] and one in a liposomal formulation [22]. This same research group also developed and evaluated Recombinant Chimeric Protein (RCP) [12], Chimera [20], and ChimT [26]. Although various studies demonstrate promising results, showing significant reductions in parasite load across different organs and assessment methods, there has been limited progress in advancing the technological development process. This includes a lack of implementation of real-world studies and/or randomized clinical trials.

Several studies have utilized bioinformatics tools for the strategic selection of immunodominant epitopes, which had been favored by the improvement of genome and proteome from *Leishmania* spp. in data banks. Among these, at least 14 studies employed in silico prediction methods to map immunogenic epitopes [12,20,23,28,30,34,35]. This approach has shown promise in rapidly identifying potential immunogenic epitopes of *Leishmania* spp. through predicted MHC binding and B cell interactions, essential steps toward inducing immune responses. While further in vitro and in vivo validation is required to confirm that these predicted epitopes effectively trigger protective immunity, this in silico methodology represents a substantial advancement, streamlining and accelerating vaccine development [46–48]. Mice, particularly the BALB/c and C57BL strains, were the primary experimental models employed in 18 analyzed studies. These models are widely used due to their well-understood immune responses, ease of handling, and availability of reagents and tools for immunological assays. However, the course of VL in mice does not always fully mirror its manifestation in humans, and different mouse strains exhibit variability in susceptibility to infection, which can complicate direct comparisons of results [49,50].

Regarding the adjuvants, except for the studies by Carcelén et al. (2009) [18] and Cotrina et al. (2018) [19], all other studies utilized adjuvants in conjunction with chimeric or multiepitope antigens. Given the relatively limited antigenic repertoire compared to inactivated or attenuated parasites, these formulations often need adjuvants to elicit a robust and long-lasting immune response [51–53]. The saponin stood out as one of the most employed in association with multiepitope antigens, showing promising results in 7/22 studies. Other adjuvants, such

as MPLA and GLA-SE, were also consistently used and demonstrated potential benefits. The selection of the most suitable adjuvant for a formulation is indeed a critical task in the development of new vaccines, as the immune response to a given adjuvant is highly dependent on the specific characteristics of the vaccine and the immunological context. Importantly various aspects of the formulation must be carefully considered when defining the adjuvant, such as administration routes, the type of immune response desired (humoral or cellular, or both), the nature of the pathogen and antigen, biological characteristics of the vaccine receptors, as well as safety and economic feasibility of the adjuvants [54]. Indeed, in many cases, when multiepitope antigens were evaluated without adjuvants, they failed to significantly reduce the parasite load in the assessed organs [12,20,24,26,27].

In the studies included in the present review, different routes were used for the challenge with *L. infantum*, especially the intravenous route [17,23,24,27,29–32,34,35,37] and the subcutaneous route [12,20–22,26,28]. Different inoculation routes, along with parasite dose, impact the course of infection and parasite load. Kaur et al. (2008) [55] demonstrated that challenge via the subcutaneous route results in a lower hepatic parasite load compared to intradermal, intraperitoneal, and intracardiac routes in terms of hepatic parasite load. In contrast, intravenous inoculation consistently results in effective infection, regardless of parasite dose, leading to persistent parasite presence in the spleen and liver of the animals [49]. Additionally, a clear dose-dependent effect of the parasites was observed. Mice inoculated subcutaneously with a low number of parasites (i.e., $10^3$) exhibited a mild infection associated with a Th1 response. In contrast, a higher number of parasites ($10^7$) resulted in a high parasite load in the spleen and lymph nodes, along with a Th2 response [56]. Intracardiac inoculation is linked to the development of a Th2 response, which facilitates the establishment of a persistent infection [55]. Parasite persistence in the spleen can be detected for at least four months post-infection [57]. Therefore, the comparison between different studies is often limited. Different parasite strains and concentrations, the routes used, and the methods employed for quantifying parasite load must all be considered [58]. Ideally, the establishment of a standardized protocol could improve the quality of studies about the development of vaccines for VL.

The most used methods for assessing parasite load were limiting dilution [12,20–22,26,29,31,32,34,35,37] and quantitative PCR (qPCR) [12,18,20,22–28,33]. Limiting dilution is a labor-intensive method that requires multiple processing steps and several days to obtain results, which significantly increases the risk of contamination. Despite these challenges, it remains a sensitive technique capable of specifically detecting live parasites, thereby confirming the presence of active disease [59–61]. In contrast, qPCR provides faster results, with kDNA was the most frequently targeted marker in these studies [12,18,20,22,24–28]. Due to its high copy number in the parasite, kDNA allows the detection of as few as $10^{-3}$ parasites in a sample [62,63]. However, the persistence of DNA in the host organism complicates the differentiation between viable and non-viable parasites [64,65]. To address this limitation, other markers have been evaluated for detecting viable *Leishmania* parasites. Recent studies suggest that spliced leader RNA (SL-RNA) degrades more rapidly than kDNA and correlates better with microscopic examination, making it a potentially superior marker for monitoring disease progression and evaluating vaccine efficacy [66,67].

The spleen is one of the primary organs affected by the *L. infantum* parasite, exhibiting persistent infection in murine models [68]. Therefore, this organ is crucial for evaluating vaccine efficacy. Generally, all included studies assessed the parasite load in the spleen, in addition to other organs. However, the studies by Duthie et al. (2017) [33] and Clímaco et al. (2023) [28], evaluated only the liver. Although both studies assessed parasite load in the same organ, there was a significant difference in the time interval between infection and outcome evaluation: 28 and 120 days, respectively. The results concerning the reduction of parasite load should be

interpreted with caution, considering the period between the challenge and the evaluation of outcomes, as the dissemination of parasites in the liver is self-limiting. In experimental infection studies, following intravenous infection with *L. infantum*, promastigotes transform into amastigotes and replicate in Kupffer cells and dendritic cells. During the first two weeks, parasite growth is uncontrolled due to an ineffective Th1 response [69]. Consequently, the parasitic load peaks between 2- and 8-weeks post-infection. After this period, hepatic infection generally resolves through the formation of granulomas, with amastigotes being almost absent [70].

Some of the included articles did not provide essential information, such as the strain of the parasite used in experimental infection [29,32], the interval between the last immunization and the challenge of the immunized animals [32], and the route of parasite administration during experimental challenge [18]. Additionally, the absence of details on methodological aspects, like the random allocation of animals to experimental groups or the blinding of researchers to the groups, compromised the ability to assess risk of bias.

## Conclusion

Based on the studies included in this review, no single vaccine formulation emerged as superior among those evaluated. However, the aim was not to compare the studies directly but to use the presented information to guide the rational development of new research involving chimeric or multiepitope antigens as active pharmaceutical ingredients for VL vaccination. Based on the findings of the reviewed studies, immunization via subcutaneous or intradermal routes were the most employed methods, alongside intravenous challenge to better mimic the course of infection, appear to be promising strategies. However, the choice of the optimal adjuvant remains highly context-dependent, varying according to the antigen and its interaction with the immune system. Furthermore, standardized protocols for evaluating immune responses and parasitic load would greatly enhance the comparability and reproducibility of results across studies. However, it is important to note that the collected data supports the feasibility of these antigens for the development of a vaccine against this disease.

## Supporting information

**S1 File. Search strategy used in each database.**
(DOCX)

## Author Contributions

**Conceptualization:** Karine Ferreira Lopes, Mariana Lourenço Freire, Edward Oliveira.

**Data curation:** Karine Ferreira Lopes, Mariana Lourenço Freire, Edward Oliveira.

**Formal analysis:** Karine Ferreira Lopes, Mariana Lourenço Freire.

**Investigation:** Karine Ferreira Lopes, Mariana Lourenço Freire.

**Methodology:** Karine Ferreira Lopes, Mariana Lourenço Freire.

**Project administration:** Edward Oliveira.

**Supervision:** Edward Oliveira.

**Validation:** Karine Ferreira Lopes, Mariana Lourenço Freire.

**Visualization:** Karine Ferreira Lopes, Mariana Lourenço Freire, Silvane Maria Fonseca Murta.

**Writing – original draft:** Karine Ferreira Lopes, Mariana Lourenço Freire, Silvane Maria Fonseca Murta, Edward Oliveira.

**Writing – review & editing:** Karine Ferreira Lopes, Mariana Lourenço Freire, Silvane Maria Fonseca Murta, Edward Oliveira.

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
