## [Decision Letter · Decision Letter 0]

17 Oct 2024

Dear Dr. Oliveira,

Thank you very much for submitting your manuscript "Efficacy of vaccines based in chimeric or multiepitope antigens for protection against visceral leishmaniasis: A systematic review" for consideration at PLOS Neglected Tropical Diseases. As with all papers reviewed by the journal, your manuscript was reviewed by members of the editorial board and by several independent reviewers. The reviewers appreciated the attention to an important topic. Based on the reviews, we are likely to accept this manuscript for publication, providing that you modify the manuscript according to the review recommendations. 

Please see comments from reviewers. Please revise manuscript accordingly.

Sincerely,

Bradford S McGwire

Guest Editor

Susan Madison-Antenucci

Section Editor

Please see comments from reviewers. Please revise manuscript accordingly.

Reviewer's Responses to Questions

**Key Review Criteria Required for Acceptance?**

**Methods**

-Are the objectives of the study clearly articulated with a clear testable hypothesis stated?

-Is the study design appropriate to address the stated objectives?

-Is the population clearly described and appropriate for the hypothesis being tested?

-Is the sample size sufficient to ensure adequate power to address the hypothesis being tested?

-Were correct statistical analysis used to support conclusions?

-Are there concerns about ethical or regulatory requirements being met?

Reviewer #1: The study presents well-articulated objectives, with a clear testable hypothesis, centered on the efficacy of vaccines based on chimeric or multiepitopic antigens for the protection against visceral leishmaniasis. The study design is adequate, using a systematic review of preclinical and clinical studies, with animal models such as rats, dogs and hamsters, which are appropriate for the hypothesis tested. The sample size seems sufficient in each individual study, although the limitation of some studies with small groups of animals may affect the generalizability of the results. Correct statistical analysis methods, such as limiting dilution and quantitative PCR, were used to support the conclusions, although the lack of details in some studies generated uncertain risk of bias. Regarding ethical requirements, the study mentions the use of the SYRCLE tool to assess the risk of bias in preclinical studies, suggesting the consideration of ethical guidelines, but without extensively detailing specific approvals for each study.

Reviewer #2: -In the collection of the characteristics of the included antigens, it is important to also collect data on from which species the original protein antigen was discovered. I.e. for the Coler et al 2007 study, it is provided as L. major stress-inducible protein (LmSTI1), but the species the antigen is discovered in was not mentioned for the other proteins. This could be important in the efficacy of a vaccine for L. infantum.

-What were the predicted MHC haplotypes for the epitopes, and what is the MHC background of the animals tested in? Do these match? This may be good information to capture as well, as they may influence vaccine efficacy.

-Do you think that including only studies assessing parasite load or positivity rate in studies involving natural infection will bias towards animal model research? Spleen parasite load assessment is a risky procedure in humans, and longitudinal natural infection assessments are often costly and therefore not performed a lot.

**Results**

-Does the analysis presented match the analysis plan?

-Are the results clearly and completely presented?

-Are the figures (Tables, Images) of sufficient quality for clarity?

Reviewer #1: The analysis presented in the study is in line with the analysis plan described, following a systematic approach to evaluate the efficacy of vaccines based on chimeric and multiepitopic antigens against visceral leishmaniasis. The results are presented clearly and completely, with a detailed description of the reductions in parasite load in different animal models. The figures and tables used are of good quality and contribute to the clarity of the data, facilitating the understanding of the results. The information is well organized, offering an accurate view of the reviewed studies, with numerical data and percentages that support the conclusions of the study.

Reviewer #2: The analysis matches the analysis plan, and the tables are clear. I have no major comment on the results aside from this. Perhaps my comments on the Methods are also part of my comments on the Results since it is a systemic review.

A minor comment: what does 'Not made' mean in the tables? Undisclosed? Please clarify.

**Conclusions**

-Are the conclusions supported by the data presented?

-Are the limitations of analysis clearly described?

-Do the authors discuss how these data can be helpful to advance our understanding of the topic under study?

-Is public health relevance addressed?

Reviewer #1: The conclusions of the study are well supported by the data presented, highlighting the potential of vaccines based on chimeric and multiepitopic antigens for protection against visceral leishmaniasis. The authors clearly discuss the limitations of the analysis, such as the risk of bias in some studies and the scarcity of clinical trials. In addition, they explore how the results can advance the development of new vaccines, highlighting the importance of additional studies to overcome the identified gaps. The public health relevance is addressed, emphasizing the need for effective vaccines to control visceral leishmaniasis, a disease that affects vulnerable populations in several regions of the world.

Reviewer #2: (No Response)

**Editorial and Data Presentation Modifications?**

Reviewer #1: No editorial or data presentation changes are necessary.

Reviewer #2: -L200: 'one antigen consisting of a pool of peptides selected through epitope prediction analyses.' is not clear to me. What is meant with this? Please clarify.

-L233-L234: It is perhaps interesting to assess what the most potent adjuvant was? 

-L66 has a typo, should be 'severe' not 'sever'.

-L68 has a typo, should be 'caused' not 'cause'

-L156 has a typo, should be 'tool' not 'toll'

-L386-L387: With 'less efficient in terms of hepatic parasite load', does this mean it decreases or increases the hepatic parasite load? Consider to clarify.

-L388: Consider to rewrite this in a 'negative' sense. So instead of 'Intravenous inoculation consistently results in effective infection', rather: 'Intravenous inoculation consistently fails to be protective' or something similar.

-What does 'Not made' mean in the tables? Undisclosed? Please clarify.

**Summary and General Comments**

Reviewer #1: The study presents a detailed and relevant systematic review on the efficacy of vaccines based on chimeric and multiepitopic antigens for protection against visceral leishmaniasis, a topic of high public health relevance, especially in endemic areas. Among the strengths, the scope of the research and the inclusion of data from preclinical and clinical studies stand out, which provides a broad view of the potential of these vaccines. In my opinion, the publication of the study is recommended.

Reviewer #2: To my knowledge, this is one of the first systematic reviews specifically focused on the efficacy of multi-epitope and chimeric vaccine antigens for visceral leishmaniasis. In my opinion, that makes it novel and interesting enough for publication. I particularly like that they aimed to not compare studies directly, but rather use the presented information to inform the public to stimulate rational development of new research into chimeric of multiepitope vaccines. However, there are some minor suggestions the manuscript could benefit from.

Major comments:

- On L82 to L84 it is written that incorporating multiepitope vaccine can increase the possibility of interaction with different mammalian MHC polymorphisms. However, compared to what do you mean? A live-attenuated or 'whole parasite' vaccine supposedly has all antigens of the parasite, so would theoretically have more possible epitopes that different MHC molecules could interact with. Similarly, a whole antigen-vaccine also has a lot of theoretical epitopes that can cover a large MHC space. Can you expand a little bit upon what you compare to and what the mechanism would be?

-On L367-L370 it is stated that the in silico prediction method has proven highly accurate in mapping immunogenic epitopes of Leishmania - I tend to disagree. These are methods that predict MHC epitopes or B cell epitopes, they do NOT predict immunogenicity. To my knowledge, there have been no validations of natural presentation of predicted MHC epitopes as of yet. While some predicted peptides may turn out 'immunogenic' in in vitro PBMC stimulations, this has not been proven to correlate with in vivo immunogenicity and protection. In fact, it has been observed in a recent cancer paper that in vitro PBMC stimulations with predicted peptides do not correspond to in vivo standards, because if the epitope is not naturally presented, there is no actual target the cognate T cells can recognize to elicit the immune response (https://doi.org/10.1016/j.jcyt.2023.12.002). I would consider to alter or remove this statement, although I welcome an open discussion on this as well if the authors want to keep it in.

-I would like to see which adjuvants were most promising.

-I would like to see the authors form a conclusion on what they think could help guide the rational development of new vaccine research: which adjuvants, what vaccination route, etc? Just a small sentence in the discussion would be fine.

-Why is this the Das et al. manuscript on the LEISHDNAVAX vaccine not included (https://doi.org/10.1126/scitranslmed.3008222)? As far as I know, they fit within the inclusion criteria and do not fall within the exclusion criteria? 

Small comments:

-L200: 'one antigen consisting of a pool of peptides selected through epitope prediction analyses.' is not clear to me. What is meant with this? Please clarify.

-L233-L234: It is perhaps interesting to assess what the most potent adjuvant was? 

-L66 has a typo, should be 'severe' not 'sever'.

-L68 has a typo, should be 'caused' not 'cause'

-L156 has a typo, should be 'tool' not 'toll'

-L386-L387: With 'less efficient in terms of hepatic parasite load', does this mean it decreases or increases the hepatic parasite load? Consider to clarify.

-L388: Consider to rewrite this in a 'negative' sense. So instead of 'Intravenous inoculation consistently results in effective infection', rather: 'Intravenous inoculation consistently fails to be protective' or something similar.

PLOS authors have the option to publish the peer review history of their article (what does this mean?). If published, this will include your full peer review and any attached files.

Reviewer #1: No

Reviewer #2: Yes: Nicky de Vrij

Figure Files:

Data Requirements:

Reproducibility:

References

---

## [Editor Report · Decision Letter 1]

4 Dec 2024

Dear Dr. Oliveira,

We are pleased to inform you that your manuscript 'Efficacy of vaccines based in chimeric or multiepitope antigens for protection against visceral leishmaniasis: A systematic review' has been provisionally accepted for publication in PLOS Neglected Tropical Diseases.

Best regards,

Bradford S McGwire

Guest Editor

Susan Madison-Antenucci

Section Editor

Shaden Kamhawi

co-Editor-in-Chief

Paul Brindley

co-Editor-in-Chief

---

## [Editor Report · Acceptance letter]

22 Dec 2024

Dear Dr. Oliveira,

We are delighted to inform you that your manuscript, "Efficacy of vaccines based in chimeric or multiepitope antigens for protection against visceral leishmaniasis: A systematic review," has been formally accepted for publication in PLOS Neglected Tropical Diseases.

Best regards,

Shaden Kamhawi

co-Editor-in-Chief

Paul Brindley

co-Editor-in-Chief
